# Dental Aerosol as a Hazard Risk for Dental Workers

**DOI:** 10.3390/ma13225109

**Published:** 2020-11-12

**Authors:** Jacek Matys, Kinga Grzech-Leśniak

**Affiliations:** 1Laser Laboratory, Oral Surgery Department, Wroclaw Medical University, 50-367 Wroclaw, Poland; jacek.matys@wp.pl; 2Department of Periodontics School of Dentistry, Virginia Commonwealth University, VCU, Richmond, VA 23298, USA

**Keywords:** biological safety, COVID-19, Er:YAG laser, high volume evacuator

## Abstract

Standard dental procedures, when using a water coolant and rotary instruments, generate aerosols with a significantly higher number of various dangerous pathogens (viruses, bacteria, and fungi). Reducing the amount of aerosols to a minimum is mandatory, especially during the new coronavirus disease, COVID-19. The study aimed to evaluate the amount of aerosol generated during standard dental procedures such as caries removal (using dental bur on a high and low-speed handpiece and Er:YAG laser), ultrasonic scaling, and tooth polishing (using silicon rubber on low-speed handpiece) combined with various suction systems. The airborne aerosols containing particles in a range of 0.3–10.0 μm were measured using the PC200 laser particle counter (Trotec GmbH, Schwerin, Germany) at three following sites, manikin, operator, and assistant mouth, respectively. The following suction systems were used to remove aerosols: saliva ejector, high volume evacuator, saliva ejector with extraoral vacuum, high volume evacuator with extraoral vacuum, Zirc^®^ evacuator (Mr.Thirsty One-Step®), and two customized high volume evacuators (white and black). The study results showed that caries removal with a high-speed handpiece and saliva ejector generates the highest amount of spray particles at each measured site. The aerosol measurement at the manikin mouth showed the highest particle amount during caries removal with the low and high-speed handpiece. The results for the new high volume evacuator (black) and the Zirc^®^ evacuator showed the lowest increase in aerosol level during caries removal with a high-speed handpiece. The Er:YAG laser used for caries removal produced the lowest aerosol amount at the manikin mouth level compared to conventional dental handpieces. Furthermore, ultrasonic scaling caused a minimal aerosol rise in terms of the caries removal with bur. The Er:YAG laser and the new wider high volume evacuators improved significantly suction efficiency during dental treatment. The use of new suction systems and the Er:YAG laser allows for the improvement of biological safety in the dental office, which is especially crucial during the COVID-19 pandemic.

## 1. Introduction

Dental professionals are considered as the workers who are most exposed to infection with the coronaviruses family, which causes the common cold to severe respiratory infections, like the Severe Acute Respiratory Syndrome Coronavirus 2 (SARS-CoV-2), in humans [1] (Figure 1). The primary transmission routes appear to be direct mucous membrane (eyes, nose, and oral cavity) contact with contagious respiratory bioaerosol. Many authors showed that during dental treatment a high amount of aerosol has been produced [2,3,4,5]. The dental aerosol with the most considerable concentration and extension is visible when using all rotary instruments (high and low-speed handpieces), air abrasion, or ultrasonic scalers. During dental treatment, sources of airborne contamination are saliva, dental instrumentation, respiratory sources, and the operative site. The potential for transmitting almost invisible aerosol and the virus infection spreading is so high; thus, dentists should recognize and minimize or even eliminate this risk within all clinical situations, especially when patients are in the incubation period, are unaware they are infected, or choose to conceal their disease.

Many procedures from the dental field like caries removal, periodontal supragingival scaling, and prosthodontics tooth preparation is associated with aerosol and splatter formation. The aerosol is composed of liquid and solid particles with a diameter of 50 μm or less, which are suspended in the air. Splatter is usually described as a mixture of air with water and/or solid substance in which water droplets are bigger than 50 μm [6]. Particular attention should be paid to eliminating aerosol particles in the range of 0.5–10 μm diameter, which can be inhaled and held on the human lung’s terminal bronchioles and alveoli [7]. The aerosol generated in the dental office during the treatment occurs in three forms. The first type of bioaerosol is contagious respiratory aerosol resulting from respiratory action, and the ballistic discharge of bronchial secretions during sneezing or coughing [4]. The second is a pure water-air mist produced by rotary instruments, scalers, piezoelectric units, or air abrasive devices. The third form is a mixture consisting of a heavily diluted aerosol spray with a respiratory bioaerosol that could potentially spread the virus for a greater distance.

The positive effect of lasers in dentistry has been proven in the scientific literature regarding different treatment modality and eliminating the virus, bacteria, and fungi [8,9,10,11,12,13,14,15,16,17,18]. Some lasers produce an aerosol or induce a bubble formation in fluids (cavitation effect) [19,20,21,22,23]. The erbium lasers are widely used for different dental procedures, including caries removal, endodontic irrigation, periodontal and implant treatment or orthodontic brackets, or prosthetic crown debonding [19,24,25,26,27]. Globally, it has been increasing awareness of the potential health risk of laser generating aerosol. Comparatively, typically respiratory droplets have a bigger size particle (usually in micrometers) than smaller viruses (in nanometers). Generally, viruses are much smaller than the cells they infect. Literature confirmed using lasers in dentistry to viruses inactivation [9,10,28,29,30]. Worth attention is the fact that laser generating heat and soft tissue vaporization, leads to smoke production, which could contain infectious particles [8,9].

Up to date, there is no evidence available on what level of the aerosolized coronavirus SARS-CoV-2 generated during dental procedures can be a source of infection. The infectious dose has not yet been determined, nor has the level of potential virulence for the enormously diluted mixed bioaerosol in dentistry been discovered. However, the virus’s ability to infect the human is directly dependent on the number of virus particles [31]. The study of Carter et al. and Seal et al. [32] found dependence in colds and other respiratory illness incidences among naval recruits and the dental professionals who treated them. Notably, the dental team needs to minimize the risk of infection during scheduled dental treatment, the methods of elimination of the third form of the aerosol (a mixture of bioaerosol and water spray) by applying the high efficient suction systems.

A systematic review of aerosol-generating procedures concerning SARS was summarized in the 2014 WHO guideline [33]. The review described the risk factors, basic properties of aerosols generated during routine dental procedures, ways in which dental aerosols are generated, and the types of pathogens they can harbor. The important conclusion from the recommendations is that dental treatment with the lowest aerosol generation is demanded and indicated. Various methods are explained in the scientific literature to reduce the hazard produced due to bioaerosols, e.g., hand protection (gloves), body protection (medical apron, caps, and footwear), eye protection (safety glasses, goggles, and face shield), face mask, ventilation system, ultraviolet filters and lamps, automated room disinfection (ARD) systems with hydrogen peroxide vapor, and high-volume evacuators (HVE) [4,7]. However, minimal data is available regarding the effectiveness of high-volume evacuators in dentistry.

The study aims to evaluate the aerosol generation during standard dental procedures, such as caries removal, dental scaling, and polishing, using the different suction systems, such as a saliva ejector and various high-volume evacuators. Additionally, the effect of an erbium laser and external vacuum on aerosol reduction was assessed.

## 2. Materials and Methods

A three-dimensional manikin model with natural tooth (number 35 in the FDI system, number 20 in the Universal system, and the American system) was used to mimic the everyday working conditions in dental offices. The airborne aerosols were measured at the selected sampling sites (1) next to the open mouth of the patient (manikin), (2) mouth position of the dental operator, and (3) mouth position of the dental assistant; Figure 2).

### 2.1. Measurement Method

Measurement of the aerosol at the assessed positions was done with the PC200 laser particle counter (Trotec GmbH, Schwerin, Germany). The sensor was placed at a distance of 2 cm, 40 cm, and 40 cm from the treatment area. The six different particle sizes (0.3 µm, 0.5 µm, 1.0 µm, 2.5 µm, 5.0 µm, and 10.0 µm) were measured during the experiment. Next, the particles in a range of 0.3–10.0 µm were summarized to receive the total amount of collected aerosol particles. The measurement at each site was repeated six times. The measurement device (The particle sensor) was turned on 60 s after each treatment was started. The time during the experiment was measured with a stopwatch. The mean results of each measurement were used for the statistical analysis (Figure 3).

### 2.2. The Office Air Standardization

The air conditioning system was turned off, and windows closed to achieve similar office air properties. Each measurement was conducted in a similar air environment at a stable number of particles in a range of 28,000–30,000. When the number of the particles in the air was out the assumed stable range, the air purifier system (NV1050, Novaerus, Ireland) with an air exchange of 800 m^3^ per hour was turned on to clean the air to the proper particles level prior to the measurements. The measurement of the air particles in the room was conducted in the central part of the office. The average time to purify the air to the demanded particle range in the office (20 m^2^) was around 5 min.

### 2.3. Suction Systems

The aerosol produced when applying different dental procedures was removed by the following suction systems/modalities: (Figure 4)

Saliva ejector EM15 (Monoart^®^ Euronda, Vicenza, Italy);High volume evacuator EM19 EVO (Monoart^®^ Euronda, Vicenza, Italy);Saliva ejector EM15 (Monoart^®^ Euronda, Vicenza, Italy) with an extraoral vacuum (MAcURAY PRO, KTMAX Inc., Seoul, Korea);High volume evacuator EM19 EVO (Monoart^®^ Euronda, Vicenza, Italy) with an extraoral vacuum (MAcURAY PRO, KTMAX Inc., Seoul, Korea);Zirc evacuator Mr. Thirsty One-step^®^ (Loser, Zirc, Buffalo, USA);Customized high volume evacuator (white)—designed and prepared by the authors;Customized high volume evacuator (black)—designed and prepared by the authors.

### 2.4. Dental Procedures

The assessment of the aerosol level was conducted during the standard dental procedures (without rubber dam) such as:Treatment of caries class I with the round diamond bur (#014) with a high-speed handpiece W&H Synea TA-98LC (W&H, Bürmoos, Austria) and all seven suction modalities (Figure 4A–G). Working parameters: 200,000 RPM (revolutions per minute), water cooling: 30 mL/min;Treatment of caries class I with the round rose bur (#014) with a low-speed handpiece W&H Synea TA-98LC (W&H, Bürmoos, Austria) and saliva ejector EM15 or high volume evacuator EM19 EVO. Working parameters: 15,000 RPM (revolutions per minute), water cooling: 30 mL/min;Treatment of caries class I with 1 mm diameter sapphire tip with a handpiece H14 of Er:YAG laser (LightWalker, Fotona, Slovenia) and saliva ejector EM15 or high volume evacuator EM19 EVO. Laser parameters: energy 300 mJ, frequency 20 Hz, power 6 W, water/air coolant 6/4;Tooth polishing with silicone rubber dental bur (Kenda AG, Vaduz, Liechtenstein) with a low-speed handpiece W&H Synea TA-98LC (W&H, Bürmoos, Austria) at 1000 and 10,000 RPM (revolutions per minute) and saliva ejector EM15 or high volume evacuator EM19 EVO. Water cooling: 30 mL/min;Dental calculus removal using ultrasound scaler HW-3H (Woodpecker Medical Instrument Co., Ltd., Guilin, China) and saliva ejector EM15 or high volume evacuator EM19 EVO at a power of 30%, water cooling: 40 mL/min.

### 2.5. Statistical Analysis

The amount of the dental aerosol during different dental procedures with a low and high-speed handpiece, scaler, and using suction devices, measured in 3 areas, was compared with the analysis of variance and post-hoc tests (multiple comparisons using the Tukey test). The statistical analysis was conducted by using Statistica software (StatSoft, Tulsa, OK, USA). Values below *p* = 0.05 were considered to be statistically significant.

## 3. Results

### 3.1. High Volume Evacuators Significantly Reduced the Aerosols *during Caries Removal with the Use of* a High-Speed Handpiece. *The* New Custom-Designed High Volume Evacuators and the Zirc^®^ Evacuator Proved to be the Most Effective

The level of the aerosol measured at the manikin mouth was significantly higher than around the dental assistant or operator’s mouth during caries removal with a high-speed handpiece and saliva ejector, saliva ejector + extraoral vacuum, high volume evacuator alone, and with the extraoral vacuum (*p* < 0.001). The exception was when the classic high-pressure evacuator with external vacuum, Zirc^®^ evacuator, and the new black and white high volume evacuators were used. The results for the new high volume evacuator (black) and the Zirc^®^ evacuator showed the lowest increase of the aerosol level in terms of the particle spray level in the air without significant differences between measured sites. Importantly, the Zirc^®^ evacuator and both new designed high volume evacuators significantly reduced aerosol levels measured at the manikin mouth and operator level in contrast to conventional suction systems (saliva ejector and high volume evacuator) alone and in combination with an extraoral vacuum (*p* < 0.001; Table 1).

### 3.2. High Volume Evacuator Significantly Reduced the Aerosols *during Caries* Removal with the Use of a Low-Speed Handpiece. The Low-Speed Handpiece Produced Less Aerosols than a High-Speed Turbine

Caries removal with the dental bur and a low-speed handpiece in combination with the saliva ejector and classic high volume evacuator showed the highest aerosol level at the manikin mouth and its significant decrease at the operator and assistant mouth, respectively (*p* < 0.001; Figure 5). Moreover, the high volume evacuator decreased the amount of aerosol at the manikin mouth and operator level approximately fourth and three times in contrast to the saliva ejector, respectively. Caries removal with a low-speed handpiece resulted in significantly lower aerosol particles in contrast to a high-speed turbine.

### 3.3. Er:YAG Laser Significantly Reduced the Aerosols *during Caries Removal as Compared with Classical Rotary Handpieces*

Er:YAG laser with classic high volume evacuator generated significantly less aerosol than the high-speed handpiece combined with a saliva ejector, saliva ejector + extraoral vacuum, high volume evacuator alone and with extraoral vacuum, and the low-speed handpiece with saliva ejector and high volume evacuator during caries removal at manikin (mean 28.6 ± 1.3), operator (mean 29.2 ± 0.8), and assistant (mean 29.0 ± 1.1) mouth levels (*p* < 0.001). The level of the aerosol measured around the manikin mouth was significantly lower than at operator mouth level during caries removal with Er:YAG laser with the saliva ejector and the high volume evacuator (*p* < 0.01). The rise in the amount of aerosol at the assistant level in contrast to the manikin was insignificant (*p* > 0.05; Figure 6).

### 3.4. Polishing the Tooth with Silicone Rubber Dental bur with a Low-Speed Contra-Angle Generated the Highest Amount of Aerosol Particles

Tooth polishing with the low-speed handpiece and silicone rubber dental bur resulted in significantly highest aerosol particles number in contrast to other treatment procedures. The level of dental spray measured around the mouth of the manikin when cleaning/polishing the tooth with silicone rubber dental bur with a low-speed contra-angle handpiece was significantly higher than that measured around the operator and assistant mouth level at 1000 and 10,000 RPM, respectively (*p* < 0.001). Increasing the RPM of the low-speed handpiece from 1000 to 10,000 with the high volume evacuator significantly reduced the amount of aerosol at the manikin’s mouth in contrast to the saliva ejector (*p* < 0.001; Table 2).

### 3.5. Ultrasonic Scaling Generated Less Aerosols then Caries Removal and Tooth Polishing Using Conventional Rotary Handpieces and Traditional Suction Systems (Saliva Ejector and High Volume Evacuator)

Ultrasonic scaling produced lower aerosols amount compared to caries removal using conventional rotary handpieces and tooth polishing with a low-speed handpiece combined with a saliva ejector and high volume evacuator (*p* < 0.001). High volume evacuator significantly decreased the aerosol level at the manikin (*p* = 0.005) and operator (*p* = 0.003) levels during the ultrasound scaling in contrast to the saliva ejector. Intragroup comparison of the level of aerosol measured at different sensor locations showed insignificant differences (*p* < 0.05; Table 3).

## 4. Discussion

Dental office personnel working in the patient’s respiratory tract are constantly exposed to potentially infectious bioaerosols [34]. Working high and low-speed handpieces pose a special risk, increasing transmissions of potentially dangerous pathogens (bacteria, viruses, and fungus) in a mixture of bioaerosol with a water spray. The results of the study showed that caries removal with a high-speed handpiece and saliva ejector generated the highest amount of spray particles at each measured site. The measurement of aerosol at the manikin mouth showed the highest particle amount during caries removal with the low and high-speed handpiece. The results for the new high volume evacuator (black) and the Zirc evacuator showed the lowest increase of aerosol level during caries removal with a high-speed handpiece. The Er:YAG laser used for caries removal produced the lowest aerosol amount at the manikin zone in contrast to both conventional dental handpieces. Furthermore, ultrasonic scaling caused a minimal aerosol rise in terms of the initial office air particle level with no differences among measured areas. The rise of aerosols when using air turbine or the ultrasonic scaler with air–water spray coolant was confirmed in various previously published studies [5,7,35,36].

A dental treatment, especially when using a high-speed turbine, generates a high amount of aerosol and splatter, possibly contaminated with bacteria, viruses, fungi, and blood [37]. Aerosols with a diameter below 50 µm remained in the office air for a prolonged time and could be inhaled by dental personnel and the patients [6]. Dental aerosol contains potentially hazardous particulate matter of submicron size smaller than 10 μm (PM10). [38,39] The more aerosol particles in the air, the higher the risk of infection is. Thus, the main target to reduce the hazard of aerosol transmission in the dental office seems to be to diminish the amount of the aerosol during common dental treatments. Conventionally the saliva ejectors are used to remove water and saliva from the oral cavity. However, these tips have a low potential of removing the aerosolized particles when using high and low speed contra-angles. The greater suction efficiency has high volume evacuators. These devices have wider tips and allow removing even 90% of particles [36]. However, the main disadvantage of the system is that it must be in very near contact with a tooth to work with high efficiency.

The main aim of the study was to answer the question: is it possible to reduce the aerosol and splatter amount produced during standard dental procedures (caries removal and ultrasonic scaling) with higher efficiency than conventional high volume evacuator and saliva ejector. The results of our study for caries removal with a high-speed contra-angle handpiece showed that wider suction systems (customized evacuators) allowed removing of 2 and 8 times more aerosols when compared with a high volume evacuators and a saliva ejector, respectively. The study of Jacks [36] showed that the high volume evacuators removed approximately 90% of aerosols during ultrasonic scaling when compared to the intraorally positioned standard saliva ejector. In our present study, the additional benefit to HVE provided extraoral evacuators, which reduced the aerosols by 33%. Additionally, better efficiency (88%) in aerosols reduction during caries removal using a high-speed turbine showed Zirc evacuator (32.1 ± 2.0) in contrast to HVE (60.5 ± 3.0). It should be mentioned that all the study was conducted without a rubber dam placement. The rubber dam may affect the amount of aerosol, but most importantly, it will limit the contact of water-air spray with mucous membranes, which increases the safety of the procedure. However, it is not always possible to place a rubber dam (not fully erupted teeth), children, people with respiratory disorders (severe asthma and breathing trouble through the nose), and special care patients.

The second important subject of the study was to assess whether ultrasonic scaling significantly increases the aerosol particles in the air of the office. Graetz et al. in their study [40] showed that ultrasonic and sonic devices are useful in eliminating biofilm but bear a greater risk to the dental workers concerning the production of splatter. Furthermore, Harrel et al. [41] observed that ultrasonic scalers, even when working without coolant, could spread the airborne material for at least 18 inches from the operative site [4]. The more considerable amount of aerosol and splatter can be formed from saliva and blood placed in the operative site. Our study results showed that if ultrasonic scaling was applied with a high volume evacuator, the increase of aerosol in terms of the mean particle amount in the office air was minimal. Less aerosol was generated during ultrasonic scaling when compared to the saliva ejector. Our study showed that aerosol produced during ultrasonic scaling was significantly less than during other procedures when using rotary instruments. Thus, ultrasonic scaling seemed to have less potential in pathogen transmission as compared to caries removal. Additionally, the high volume evacuator allowed the remaining aerosol to be similar to this level at the starting point of the experiment (2800–3000 particles).

The important issue was to assess the aerosol level when the Er:YAG laser was used for caries removal. The Er:YAG laser with the wavelength of 2940 um has the highest absorption in the water; hence it can be used to remove both hard and soft tissue [16,24,25,26,27]. In the study, we used the Er:YAG laser at recommending by manufacturer settings (energy 300 mJ, frequency 20 Hz, power 6 W, and water/air coolant 6/4). An important finding of the study was that the Er:YAG laser with a high volume evacuator generated significantly less aerosol than the high-speed handpiece combined with a saliva ejector, saliva ejector + extraoral vacuum, high volume evacuator alone and with extraoral vacuum, and the low-speed handpiece with saliva ejector and high volume evacuator during caries removal at manikin (mean 28.6 ± 1.3), operator (mean 29.2 ± 0.8), and assistant (mean 29.0 ± 1.1) mouth levels. Contrary to our aerosol generation results by Er:YAG laser, Guderian et al. found excessive aerosol release during CO_2_ laser application. That can indicate additional benefit in using erbium family lasers compared to CO_2_ in terms of decreasing biohazard in the dental office [42]. Although the laser works through the water components vaporization in the target tissue, the explosion effect has a limited range; thus, the aerosol formed during this process is lower in contrast to traditional methods of caries removal [21]. This is a possible explanation of lower aerosol formation obtained during caries removal with the Er:YAG laser. Furthermore, the interaction of the laser light with the target tissue also could have some additional benefit in the reduction of the risk of infection in dentistry.

In dental offices, rotary devices produce a large amount of aerosols containing parts of the tooth, saliva, blood, and many dangerous micropathogens during operation. One way to reduce the risk of infection with the SARS-CoV-2 virus was to close dental offices or admit patients on an emergency basis. However, this limited patient availability to dental services could also have the potential adverse effect of overuse of painkillers [43]. Many methods of disinfecting surfaces and air in dental offices (ozonation and fumigation with hydrogen peroxide) to maintain biological safety in the office are proposed [34]. The World Dental Federation (FDI) also recommended using Type FFP2 and FFP3 masks and a full protective apron while working [33]. Based on our findings presented in the paper, it is also essential to reduce the amount of aerosols during the dentist’s work by using highly efficient suction systems and modern devices that generate less aerosols during their operation. Our present study showed a significantly lower formation of aerosols during caries’ removal with the Er:YAG laser. The disadvantage of this method is the high cost of the device itself. However, wider dental high volume evacuators, profiled in such a way that partially embraced the head of a classic dental handpiece, allows an inexpensive and efficient reduction of aerosols when working in the oral cavity. An additional advantage of this solution is the possibility of the operator working without assistance because the wider high volume evacuator’s back surface also supports buccal tissues in the treatment area. However, additional studies should be done to check the virus inactivation potential during caries removal with the Er:YAG laser. A future great perspective for the laser light and potential work with the safety recommendation like being good in size and the shape of high-volume evacuators is promising, particularly in the current COVID-19 pandemic.

## 5. Conclusions

The most intensive dental aerosol is presenting during work with a high-speed handpiece or ultrasound scaler. We are obliged to reduce the probability of infection transmission during dental procedures and preserve all dental office workers and patients. The results showed that the amount of aerosol generated during caries removal, tooth polishing, and ultrasonic scaling was strictly connected with the method/device used in each procedure and significantly with the suction device. The main finding of the study confirmed that high volume evacuators allowed removing a significant amount of aerosol. The highest efficiency in aerosol reduction was obtained for wider customized high-volume evacuators. Additionally, the Er:YAG laser utilized for caries removal was characterized by a low aerosol generation even when working combined with saliva ejector. We do not recommend to use saliva ejector alone during tooth polishing with silicone rubber dental bur, ultrasonic scaling, and when treating caries conventionally.

## Figures and Tables

**Figure 1 materials-13-05109-f001:**
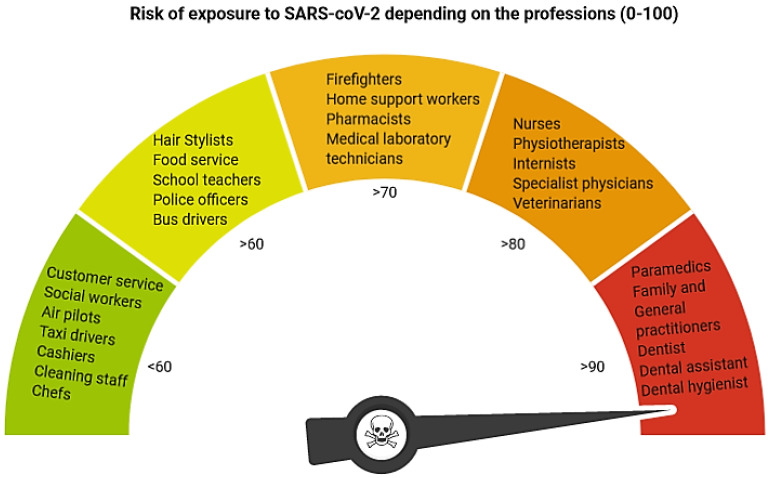
Risk of exposure to SARS-CoV-2 based on the Alberta Federation of Labour (created in BioRender.com).

**Figure 2 materials-13-05109-f002:**
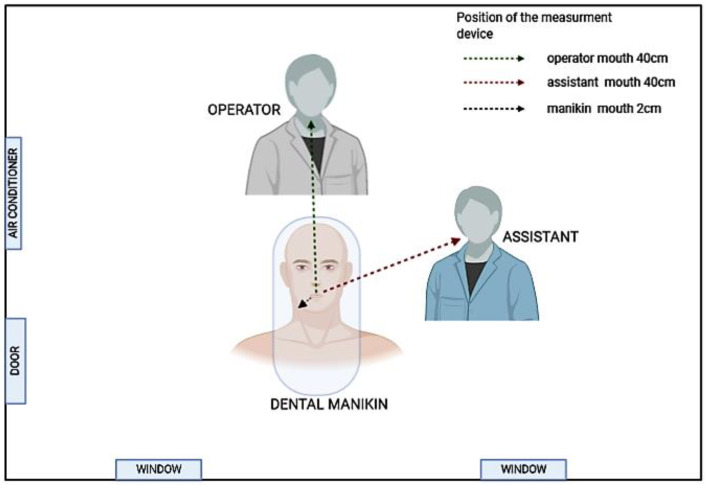
Measurement sites and positions of the sensor.

**Figure 3 materials-13-05109-f003:**
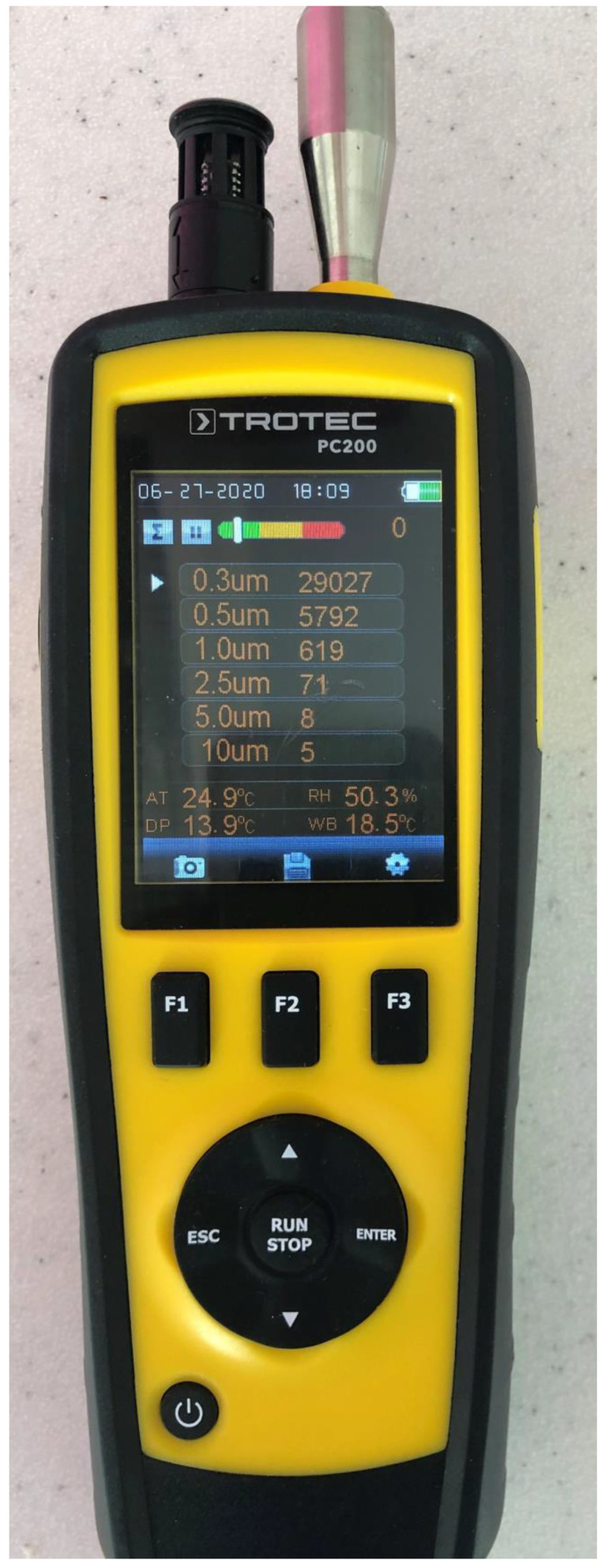
Particle sensor PC200 (Trotec GmbH, Schwerin, Germany) used in the experiment.

**Figure 4 materials-13-05109-f004:**
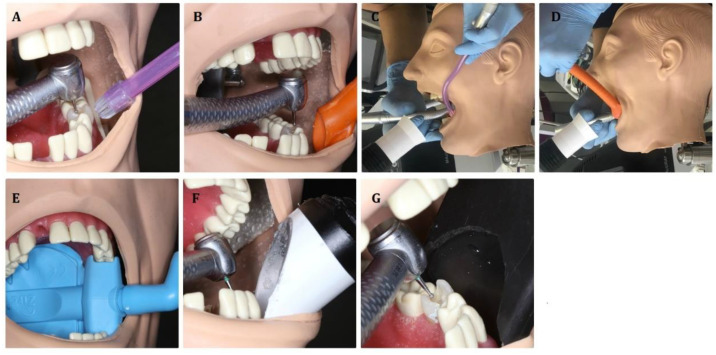
The suction systems used in the study. (**A**) Saliva ejector. (**B**) High volume evacuator. (**C**) Saliva ejector with an extraoral vacuum. (**D**) High volume evacuator with an extraoral vacuum. (**E**) Zirc^®^ evacuator. (**F**) Customized high volume evacuator (white). (**G**) Customized high volume evacuator (black).

**Figure 5 materials-13-05109-f005:**
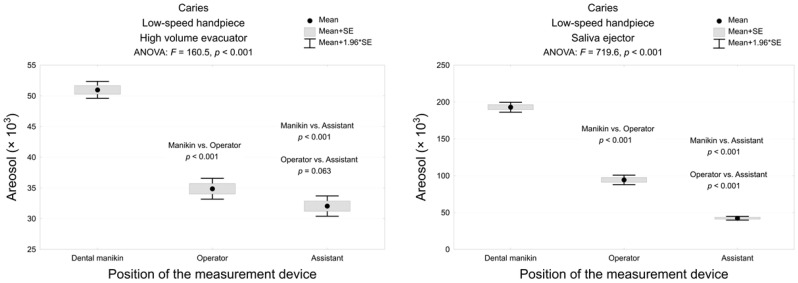
The amount of the dental spray during caries removal with low-speed handpiece with a high volume evacuator or saliva ejector, measured in 3 areas and the analysis of variance and post-hoc test (multiple comparisons using the Tukey test).

**Figure 6 materials-13-05109-f006:**
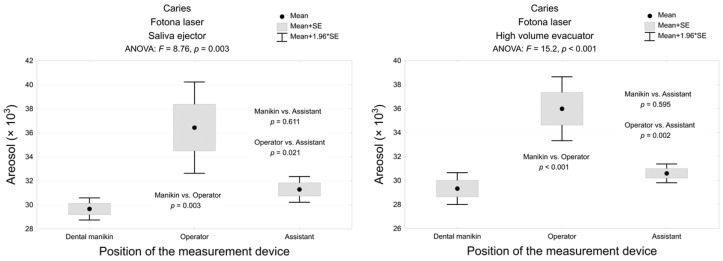
The amount of the dental spray during caries removal with Er:YAG laser with a high volume evacuator or saliva ejector, measured in 3 areas and the analysis of variance and post-hoc test (multiple comparisons using the Tukey test).

**Table 1 materials-13-05109-t001:** The amount of aerosol (×10^3^) measured at the three sites (mean ± SD) during caries removal using the high-speed handpiece with; A. Saliva ejector. B. High volume evacuator. C. Saliva ejector with an extraoral vacuum. D. High volume evacuator with an extraoral vacuum. E. Zirc^®^ evacuator. F. Customized high volume evacuator (white). G. Customized high volume evacuator (black).

Procedure	Tools	Exhaustion	Place of Measurement			ANOVA *p*
			Manikin (A)	Operator (B)	Assistant (C)	
Caries	High-speed handpiece	Saliva ejector	260.2 ± 20.6	121.2 ± 20.8	121.5 ± 9.5	A vs. BC *p* < 0.001
		Saliva ejector + extraoral vacuum	120.3 ± 18.3	43.9 ± 2.6	35.1 ± 1.6	A vs. BC *p* < 0.001
		High volume evacuator	60.5 ± 3.0	40.7 ± 3.3	31.5 ± 2.4	A vs. BC *p* < 0.001 B vs. C p < 0.001
		High volume evacuator + extraoral vacuum	40.7 ± 0.9	43.9 ± 1.9	44.5 ± 1.8	A vs. BC *p* < 0.01
		Zirc evacuator	32.1 ± 2.0	31.0 ± 1.4	31.0 ± 1.0	0.353
		New high volume evacuator—black	31.6 ± 1.1	31.5 ± 1.1	31.5 ± 0.8	0.979
		New high volume evacuator—white	31.5 ± 0.9	34.0 ± 1.2	31.2 ± 1.0	B vs. AC *p* < 0.001

**Table 2 materials-13-05109-t002:** The amount of the dental spray during tooth’s polishing measured in 3 areas and the analysis of variance and post-hoc test (multiple comparisons using the Tukey test).

Place of Measurement	Tooth Polishing Procedure with the Low-Speed Handpiece					
	1000 rpm			10000 rpm		
	Saliva Ejector	High Volume Evacuator	*p*-Value	Saliva Ejector	High Volume Evacuator	*p*-Value
Manikin	345.7 ± 65.8	201.0 ± 13.9	<0.001	330.1 ± 40.3	79.7 ± 3.1	<0.001
Operator	37.9 ± 2.3	34.6 ± 3.1	0.063	35.3 ± 6.9	34.5 ± 1.9	0.814
Assistant	31.6 ± 1.0	31.8 ± 1.5	0.803	32.7 ± 2.8	30.9 ± 1.3	0.192
	<0.001	<0.001		<0.001	<0.001	

**Table 3 materials-13-05109-t003:** The amount of the dental spray during ultrasonic scaling with a high volume evacuator or saliva ejector, measured in 3 areas and the analysis of variance and post-hoc test (multiple comparisons using the Tukey test).

Place of Measurement	Exhaustion	Test Result
Saliva Ejector	High Volume Evacuator
Manikin	34.2 ± 2.8	29.7 ± 1.4	0.005
Operator	32.5 ± 2.3	28.4 ± 1.2	0.003
Assistant	31.0 ± 1.6	29.6 ± 0.6	0.079
ANOVA *p*	0.072	0.139

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
