# Peer review of "Dental Aerosol as a Hazard Risk for Dental Workers"

_materials, 2020, doi:10.3390/ma13225109_

Round 1

Reviewer 1 Report

I read very interesting and novel research article. The aim of the study is novel, impressive and beneficial for the dental society.

The manuscript is novel, scientifically sound and well written, thus can be accepted after some minor corrections.

1) In abstract write the importance of study particularly how it can be related with COVID.

2) In abstract mention the values of results of the study. Add keywords related to study.

3) Unit style should be in standard format,  give full form of used abbreviation somewhere in the manuscript.

4) Still more discussion on practical applications and future perspectives of the study is essential.

Author Response

Reviewer: 1

Dear reviewer, thank you for your great effort in review of our manuscript.

Comments to the Author

I read very interesting and novel research article. The aim of the study is novel, impressive and beneficial for the dental society. The manuscript is novel, scientifically sound and well written, thus can be accepted after some minor corrections.

1) In abstract write the importance of study particularly how it can be related with COVID.

2) In abstract mention the values of results of the study. Add keywords related to study.

Ad.1 and 2 We underlined the importance of the findings in terms of COVID-119 in the abstract and we add a new keyword.

3) Unit style should be in standard format, give full form of used abbreviation somewhere in the manuscript.

Ad.3 Description of all the abbreviations in the text of the paper was added.

4) Still more discussion on practical applications and future perspectives of the study is essential.

Ad.4 We added a new paragraph to “Discussion” describing the practical applications of the findings presented in the paper.

All the changes are highlighted in blue color.

Reviewer 2 Report

The authors submitted a manuscript with results of a study with a big potential. However, the overall merit of the text is weekend by the presentation of the results, which should be more straightforward to clearly support the conclusions.

Specific comments:

-It is not specified if the rubber dam was used during the treatment procedures. It could have affected the results and should be mentioned.

-The presentation of the results is confusing. Different types of tables/charts are used among the five results sections. In first section the results for all ways of exhaustion are given, whereas in all other section only for two of them. From the practical point of view, the most interesting results are (i) the comparison of different treatment modalities in production of aerosol and (ii) the comparison of the effectivity of the different ways of exhaustion. These results are a bit hidden and the reader must laboriously extract them from the presented results. In spite of this, these desired results are discussed and mentioned in the Conclusions. Which makes it even more important to present them in a more suitable way in the Results section.

-Line 341-2: I do not understand what this sentence means in terms of the rubber dam. This is the only place where the rubber dam is mentioned and its (non)recommending is not supported by the results.

Minor issues:

-Lines 48, 62, 90-91 and many more times throughout the text: We do not actually treat the caries. We treat the tooth and remove the caries. Please change to "caries removal".

-Line 53: Remove the dash after 0.5, as the "to" follows.

-Line 58: Remove one of "last" and "third", they say the same.

-Line 60: It is not clear from the formulation whether you men that the laser is recommended for the treatment of the infections, or to for the dental treatment when these infections are present. Please rephrase.

-Line 107: There should be spaces between the numbers and "cm".

-Line 127: Pleas specify the time for the volume 800 m3. Per minute? Hour?

-Line 193: Here it is necessary to write 3 as the upper index. "× 103" is wrong. Several times in the text. I appreciate using of multiplication sign in stead of commonly used letter x.

Author Response

Reviewer: 2

Dear reviewer, thank you for your great effort in review of our manuscript.

Comments to the Author
The authors submitted a manuscript with results of a study with a big potential. However, the overall merit of the text is weekend by the presentation of the results, which should be more straightforward to clearly support the conclusions.

Specific comments:

-It is not specified if the rubber dam was used during the treatment procedures. It could have affected the results and should be mentioned.

Ad.1 We do not used a rubber dam in the study. We shortly explained why in discussion. We also added this information to the M&M section.

-The presentation of the results is confusing. Different types of tables/charts are used among the five results sections. In first section the results for all ways of exhaustion are given, whereas in all other section only for two of them. From the practical point of view, the most interesting results are (i) the comparison of different treatment modalities in production of aerosol and (ii) the comparison of the effectivity of the different ways of exhaustion. These results are a bit hidden and the reader must laboriously extract them from the presented results. In spite of this, these desired results are discussed and mentioned in the Conclusions. Which makes it even more important to present them in a more suitable way in the Results section.

Ad.2 Thank you for these essential comments. The findings included in 5 subsections of the Results were extracted and highlighted in each subsection's titles. Also, we added important issues to the text of the result to present the findings in a way easier to read.

-Line 341-2: I do not understand what this sentence means in terms of the rubber dam. This is the only place where the rubber dam is mentioned and its (non)recommending is not supported by the results.

Ad.3 Incorrect sentence was corrected. We do not used a rubber dam in the study. We shortly explained why in discussion.

Minor issues:

-Lines 48, 62, 90-91 and many more times throughout the text: We do not actually treat the caries. We treat the tooth and remove the caries. Please change to "caries removal".

Ad.4 We changed „caries treatment“ to “caries removal“ in whole text of the paper.

-Line 53: Remove the dash after 0.5, as the "to" follows.

Ad.5 Incorrect sentence was corrected.

-Line 58: Remove one of "last" and "third", they say the same.

Ad.6 Incorrect sentence was corrected.

-Line 60: It is not clear from the formulation whether you men that the laser is recommended for the treatment of the infections, or to for the dental treatment when these infections are present. Please rephrase.

Ad.7 Lasers can be useful with for different type of infections treatment, thus we changed the sentence you mentioned to “The positive effect of lasers in dentistry has been proven in the scientific literature regarding different treatment modality and eliminating the virus, bacteria, and fungi”

-Line 107: There should be spaces between the numbers and "cm".

Ad.8 The space was added.

-Line 127: Please specify the time for the volume 800 m3. Per minute? Hour?

Ad.9 We added the correct unit – 800 m3 per hour

-Line 193: Here it is necessary to write 3 as the upper index. "× 103" is wrong. Several times in the text. I appreciate using of multiplication sign instead of commonly used letter x.

Ad.10 Upper indexes were added.

The changes are highlighted in blue color.

Reviewer 3 Report

Dental aerosol as a hazard risk for dental workers

Dear author, congratulation for the paper, but I have some issues that need clarification.

----

The study aimed to evaluate the amount of aerosol generated during standard dental procedures, combined with various suction systems.

Abstract

Line 9. “Abstract: w Standard…” Remove w.

Line 14-15. “The airborne aerosols containing particles in a range of 0.3 - 10.0 mm”. Did you mean 0.3-10.0 µm?

  1. Materials and Methods

Line 98. The airborne aerosols were measured at the selected sampling sites… How many times did you made mesurements?

The aerosol produced when applying different dental procedures was removed by the following 7 suction systems.

The assessment of the aerosol level was conducted during 5 standard dental procedures.

  1. Results

Line 183.”…extraoral vacuum. (p < 0,001)”. Correct to: “…extraoral vacuum (p < 0,001).”

Line 192. Table 1. Rephrase.

The results presented in the Table 1 and in the Figure 5 are the same. Remove one of them.

Line 209. Figure 6. Represent the amount of the dental spray during caries treatment with low-speed handpiece with a high volume evacuator or saliva ejector, measured in 3 areas. Why did you not presente the results of the other 5 suction systems?

Line 224. Figure 7. Represente the amount of the dental spray during caries treatment with Er:YAG laser with a high volume evacuator or saliva ejector. Why did you not presente the results of the other 5 suction systems?

Line 238 and 241.Table 2 and table 3 presente the same results and with a high volume evacuator or saliva ejector.

Why did you presente 2 tables with the same results and Why did you not presente the results of the other 5 suction systems?

Line 253. Represent the amount of the dental spray during ultrasonic scaling with a high volume evacuator or saliva ejector, measured in 3 areas. Why did you not presente the results of the other 5 suction systems?

Line 314-317. This sentence is not according the results!

Author Response

Reviewer: 3

Dear reviewer, thank you for your great effort in review of our manuscript.

Comments to the Author
Dear author, congratulation for the paper, but I have some issues that need clarification.

The study aimed to evaluate the amount of aerosol generated during standard dental procedures, combined with various suction systems.

Abstract

Line 9. “Abstract: w Standard…” Remove w.

Ad.1 This was corrected.

Line 14-15. “The airborne aerosols containing particles in a range of 0.3 - 10.0 mm”. Did you mean 0.3-10.0 µm?

Ad.2 This was corrected. Should be μm.

  1. Materials and Methods

Line 98. The airborne aerosols were measured at the selected sampling sites… How many times did you made mesurements?

Ad.3 The measurement at each site was repeated six times. This is stated in „Measurement methods.“

The aerosol produced when applying different dental procedures was removed by the following 7 suction systems. The assessment of the aerosol level was conducted during 5 standard dental procedures.

Ad.4 In the study, we used 7 different suction systems/modalities (saliva ejector, classic high-volume evacuator, external evacuator+saliva ejector, external evacuator+classic high volume evacuator, Zirc evacuator and 2 custom-made evacuators). Following 5 procedures were tested: caries removal with a dental turbine, caries removal with the low-speed handpiece, caries removal with Er:YAG laser, ultrasonic scaling, and tooth polishing.

Results

Line 183.”…extraoral vacuum. (p < 0,001)”. Correct to: “…extraoral vacuum (p < 0,001).”

Ad.5 The dot was removed.

Line 192. Table 1. Rephrase.

Ad.4 We changed Table 1 title to “The amount of aerosol (× 103) measured at the three sites (mean ± SD) during caries removal using the high-speed handpiece with; A. Saliva ejector. B. High volume evacuator. C. saliva ejector with an extraoral vacuum. D. High volume evacuator with an extraoral vacuum. E. Zirc® evacuator. F. Customized high volume evacuator (white). G. Customized high volume evacuator (black).”

The results presented in the Table 1 and in the Figure 5 are the same. Remove one of them.

Ad.6 Figure 5 was removed.

Line 209. Figure 6. Represent the amount of the dental spray during caries treatment with low-speed handpiece with a high volume evacuator or saliva ejector, measured in 3 areas. Why did you not presente the results of the other 5 suction systems?

Line 224. Figure 7. Represente the amount of the dental spray during caries treatment with Er:YAG laser with a high volume evacuator or saliva ejector. Why did you not presente the results of the other 5 suction systems?

Line 253. Represent the amount of the dental spray during ultrasonic scaling with a high volume evacuator or saliva ejector, measured in 3 areas. Why did you not presente the results of the other 5 suction systems?

Ad.7 Caries removal with the dental bur and a low-speed handpiece was done in combination with the salivary ejector and classic high volume evacuator only. We tested all the suction systems only for caries treatment using a dental turbine, which, according to the research, causes the greatest increases in aerosols. However, we also will test other suction systems and modalities for caries treatment using a low-speed handpiece in the near future. We added a more deep description of what suction system was used in M&M “Dental procedures”

Line 238 and 241.Table 2 and table 3 presente the same results and with a high volume evacuator or saliva ejector.

Why did you presente 2 tables with the same results and Why did you not presente the results of the other 5 suction systems?

Ad.8 Table 3 was removed.

Line 314-317. This sentence is not according the results!

Ad.9 This sentence was inserted in Discussion based on the results.: “the Er:YAG laser with classic high volume evacuator  generated significantly less aerosol than the high-speed handpiece combined with a saliva ejector, saliva ejector+extraoral vacuum, high volume evacuator alone and with extraoral vacuum, and the low-speed handpiece with saliva ejector and high volume evacuator during caries removal at manikin (mean 28.6±1.3), operator (mean 29.2±0.8) and assistant (mean 29.0±1.1) mouth levels.”

The changes are highlighted in blue color.

Round 2

Reviewer 3 Report

Dear author,

Thanks for  the new version considering the review comments.

Best regards,